# Distally Prophylactic Lymphaticovenular Anastomoses after Axillary or Inguinal Complete Lymph Node Dissection Followed by Radiotherapy: A Case Series

**DOI:** 10.3390/medicina58020207

**Published:** 2022-01-29

**Authors:** Diletta Maria Pierazzi, Sergio Arleo, Gianpaolo Faini

**Affiliations:** 1Department of Medicine, Surgery and Neuroscience, Division of Plastic and Reconstructive Surgery, “Santa Maria alle Scotte” Hospital, University of Siena, 53100 Siena, Italy; 2Department of Plastic Surgery, ASST Spedali Civili di Brescia, 25121 Brescia, Italy; aquarion5@gmail.com; 3Operative Unit of Reconstructive Microsurgery, Department of Surgery, ASST Valle Camonica, 25040 Brescia, Italy; gianpaolofaini@gmail.com

**Keywords:** lymphedema, ICG lymphography, lymphatic surgery, preventive lymphaticovenular anastomosis, supermicrosurgery

## Abstract

*Background and Objectives*: Lymphedema is an important and underestimated condition, and this progressive chronic disease has serious implications on patients’ quality of life. The main goal of research would be to prevent lymphedema, instead of curing it. Patients receiving radiotherapy after lymph node dissection have a significantly higher risk of developing lymphedema. Through the prophylactic use of microsurgical lymphaticovenular anastomoses in selected patients, we could prevent the development of lymphedema. *Materials and Methods*: Six patients who underwent prophylactic lymphaticovenular anastomoses in a distal site to the axillary or groin region after axillary or inguinal complete lymph node dissection followed by radiotherapy were analyzed. Patients characteristics, comorbidities, operative details, postoperative complications and follow-up assessments were recorded. *Results*: Neither early nor late generic surgical complications were reported. We observed no lymphedema development throughout the post-surgical follow-up. In particular, we observed no increase in limb diameter measured at 1, 3, 6 and 12 months postoperatively. *Conclusion*: In our experience, performing LVA after axillary or groin lymphadenectomy and after adjuvant radiotherapy, and distally to the irradiated area, allows us to ensure the long-term patency of anastomoses in order to obtain the best results in terms of reducing the risk of iatrogenic lymphedema. This preliminary report is encouraging, and the adoption of our approach should be considered in selected patients.

## 1. Introduction

The lymphatic system plays an important homeostatic role in the immune system, lipid metabolism and fluid balance [1]. The obstruction of the lymphatic flow, clinically manifested as lymphedema, can be caused by various conditions, such as infections, congenital malformations, traumatic injuries, surgical intervention sequelae or malignancies [2].

Secondary lymphedema, excluding the hereditary one, is an important and underestimated condition; this progressive chronic disease has serious implications on patients’ quality of life. It is often misdiagnosed, and it is frequently associated with severe morbidity and disability, affecting daily activities and causing long-term health, functional, aesthetic and economic impacts [3]. Patients are prone to develop recurrent infections, which may show as cellulitis, erysipela or lymphangitis, often requiring hospitalization [4]. The management of lymphedema is historically focused on a conservative approach that includes physical therapy, such as manual lymphatic drainage and compression garments. However, lymphatic surgery with microsurgical lymphaticovenular anastomoses (LVA) and lymph node transfers are growing in importance as surgical means to improve lymphedema treatment, but long-term improvement is obtained in a low percentage of patients only [5,6].

The main goal of research would be to prevent lymphedema instead of curing it. Through the prophylactic use of microsurgical lymphaticovenular anastomoses in patients at high risk of developing lymphedema, that goal could be achieved. Patients receiving radiotherapy after lymph node dissection have a significantly higher risk of developing lymphedema [7,8,9,10].

Lymphaticovenular bypass is a microsurgical technique that reroutes lymphatic fluid into the venous system via microsurgical anastomoses between functional lymphatic vessels and nearby low-pressure veins distally to the obstruction to the lymphatic flow [11]. This technique was introduced as a treatment to potentially reverse the progression of lymphedema; however, nowadays, it is emerging as a preventive microsurgical procedure with the aim of reducing the risk of lymphedema development in selected patients [12,13].

The purpose of our study was to investigate the efficacy of performing prophylactic lymphaticovenular anastomoses after axillary or inguinal complete lymph node dissection followed by radiotherapy. LVAs were performed distally to the site of lymph node dissection in order to obtain the best results in terms of reducing the risk of iatrogenic lymphedema.

## 2. Materials and Methods

We performed a prospective analysis to evaluate patients who underwent prophylactic lymphaticovenular anastomoses distally to the axillary or groin region to prevent limb lymphedema at our institution. Inclusion criteria: patients who have undergone axillary or inguinal complete lymph node dissection followed by radiotherapy, and who later underwent preventive lymphaticovenular anastomoses. The study was conducted in accordance with the Helsinki Declaration of 1964 (revised in 2008). Written informed consent was obtained from all the patients who underwent this surgical procedure. Exclusion criteria: patients affected by other conditions or prior surgery, which might have compromised the lymphatic drainage of the primary site.

Patients characteristics, comorbidities, operative details, postoperative complications and follow-up were recorded (Table 1 and Table 2). All patients have undergone a monthly clinical follow-up throughout the first year and then every 6 months. The upper or lower limb circumference was measured every 10 cm from the wrist or the ankle, at each follow-up check, to detect lymphedema relapse, defined as an increase in the circumference of the fixed points measured along the limb [14]. For patients with subjective symptoms of lymphedema or other signs on physical examination, changes in follow-up circumferential measurements were recorded.

### Surgical Technique

Lymphaticovenular anastomoses were performed following axillary or inguinal complete lymph node dissection. We decided to perform this treatment after the completion of adjuvant radiotherapy. All the procedures were performed under general anesthesia. Antibiotic prophylaxis was administered before surgery. The lymphatic ducts were mapped before surgery with indocyanine green (ICG) fluorescent imaging, following a distal-to-proximal sequential injection technique; a total of 2 cc of ICG was injected into the second and third interdigital space [15]. Three (for lower limb) or four (for upper limb) skin incision lines of approximately 3 cm in length near the lymphatic vessels were marked along the limb. After skin incisions, under microscope magnification (20–30×), lymph ducts were identified, and their proximal ends were ligated. Nearby subdermal veins with an adequate caliber, which should be less than 0.8 mm if performing termino-terminal anastomosis, were identified, and their distal ends were ligated. The anastomoses were, thus, performed between the proximal end of the vein and the distal end of the lymphatic duct at each preoperatively identified site. The microsurgical technique is the same standard technique for the LVA procedure [16]. Lymphatic vessels were sutured to the vein end to end with single stitches (nylon 11-0), and the anastomosis was checked for patency; we routinely confirmed patency in all cases with ICG lymphangiography (Figure 1 and Figure 2). The procedure typically lasted approximately two hours. At the conclusion of the procedure, the treated limb was immediately compressed with a short-stretch compression bandage and successively with compression garments for 3 months.

## 3. Results

From June 2020 to July 2021, six patients underwent LVA surgery. Follow-ups at a maximum of 16 months have been conducted, with minimum of 3 months still ongoing. The time between lymph node dissection and LVA was a minimum of 85 days, and a maximum of 130 days, with a median of 1085 days. All of the procedures were well tolerated by the patients. Neither early nor late generic surgical complications were reported, such as infection, wound dehiscence, skin necrosis, bleeding, hematoma, seroma and lymphocele.

We observed no lymphedema development throughout the post-surgical follow-up, and in particular we observed no increase in limb diameter measured at 1, 3, 6 and 12 months postoperatively. The hospitalization length was 2–3 days. All the patients returned to their pre-surgical level of activity.

## 4. Discussion

Secondary lymphedema is a feared complication that sometimes occurs after the surgical and radiotherapy treatment of cancer: patients who require procedures that adversely affects lymphatic function are at significant risk of developing lymphedema [17]. The management of lymphedema has historically focused on conservative measures, but a growing amount of evidence supports the effectiveness of surgical techniques in improving the long-term disability and functional impairment inflicted by lymphedema [18,19]. Microsurgical techniques allowed the development of a series of surgical options for the treatment of lymphedema, categorized as physiologic procedures [20,21,22]. Among these, one of the most commonly practiced procedures is lymphaticovenous anastomosis, a surgical approach aimed at bypassing the areas of damaged lymphatics by diverting lymph flow into the venous system prior to the areas of obstruction. The lymphovenous bypass procedure involves the identification of lymphatic vessels and bypassing these into neighboring venules [23,24]. Another microsurgical technique is the vascularized lymph node transplant procedure, which involves the microvascular anastomosis of functional lymph nodes into an extremity, either to an anatomical (orthotopic) or non-anatomical (heterotopic) location, to restore physiologic lymphatic function; several lymph node flaps have been described, including the groin flap, submental flap, supraclavicular flap, gastroepiploic flap and thoracic lymph node flap [25,26]. Suction-assisted lipectomy and direct excisional procedures are other surgical techniques for lymphedema treatment, especially for chronic lymphedema, which is characterized by fibroadipose soft-tissue deposition that can be removed by lipectomy, either minimally invasively using liposuction, or by direct excision [2].

Nevertheless, the target should be to perform prophylactic surgical procedures in order to prevent lymphedema development following axillary or inguinal complete lymph node dissection and adjuvant radiotherapy [27]. Lymphedema after standard axillary lymph node dissection can occur in up to approximately 50% of patients, and adjuvant radiotherapy to the breast or lymph nodes increases the risk of lymphedema [28,29]. The use of microsurgical procedures to prevent lymphedema was introduced in 2008 by Boccardo and et al. [30]. The LYMPHA (“lymphatic microsurgical preventive healing approach”) procedure consists of performing preventive LVA between the arm lymphatics and collateral branches of the axillary vein at the same time as nodal dissection in order to reduce postoperative lymphedema, improving patients’ quality of life [13,31]. Immediate lymphatic reconstruction is a method which may decrease the risk of lymphedema development to 6.6% [32]. Moreover, for preventing or reducing the occurrence of lower limb lymphedema, several authors have reported the use of prophylactic LVA at the same time as ilioinguinal nodal dissection [33,34,35].

Chen et al. proposed a different approach for performing prophylactic, delayed, distally based LVA in the elbow and forearm 1–2 weeks after axillary lymph node dissection [36]. The rationale of this procedure is to avoid draining afferent vessels associated with cancer-containing lymph nodes into systemic circulation and to prevent a nonfunctioning LVA due to proximal lymph-vein pressure gradients with small lymph vessels and large veins.

In our experience, the rationale of performing LVA after adjuvant radiotherapy, and distally to the irradiated area, allows us to ensure the long-term patency of anastomoses, avoiding the local radiation of adverse effects on tissues. Furthermore, our choice to perform delayed LVA, distally to the site of the lymph node dissection, has the purpose of avoiding their possible closure due to the scarring process that takes place in the axillary and groin region after lymphadenectomy. Unfortunately, there are some limitations to our study: the number of cases is small and the observation period is short. Furthermore, there is no consensus about the period of follow-up at which to determine the success of prophylactic LVAs. In addition, there are many factors that affect lymphatic function, such as age and obesity. In fact, lymphatic contraction force is decreased in older patients, and weight gain can be a concomitant cause of lymphedema worsening [37,38,39]. Therefore, it is difficult to determine the success rate of prophylactic LVAs, especially in a short follow-up, so lifetime observation is recommended for this kind of patients, but our preliminary results are encouraging.

## 5. Conclusions

We performed LVAs in patients undergoing axillary or groin lymph node dissection and radiotherapy to prevent upper/lower extremity lymphedema with highly favorable preliminary results. A longer follow-up and a larger sample are required to determine the efficacy of this approach when compared with traditional immediate prophylactic LVA.

This preliminary report is encouraging, and the adoption of our approach should be considered in selected patients.

## Figures and Tables

**Figure 1 medicina-58-00207-f001:**
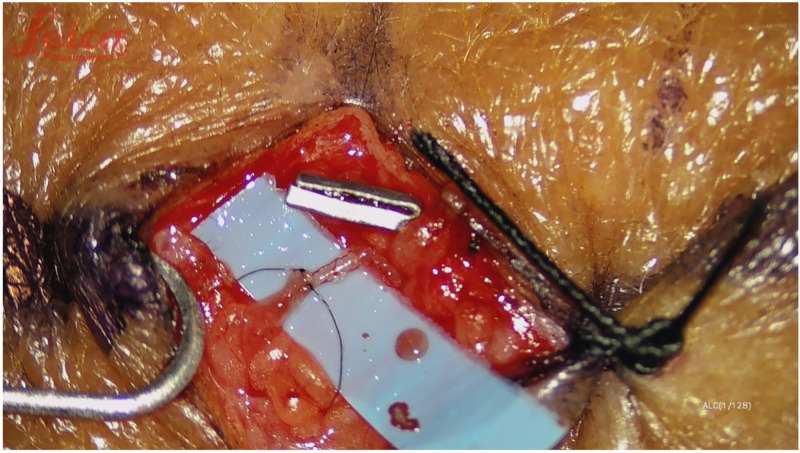
End to end lymphaticovenous bypass.

**Figure 2 medicina-58-00207-f002:**
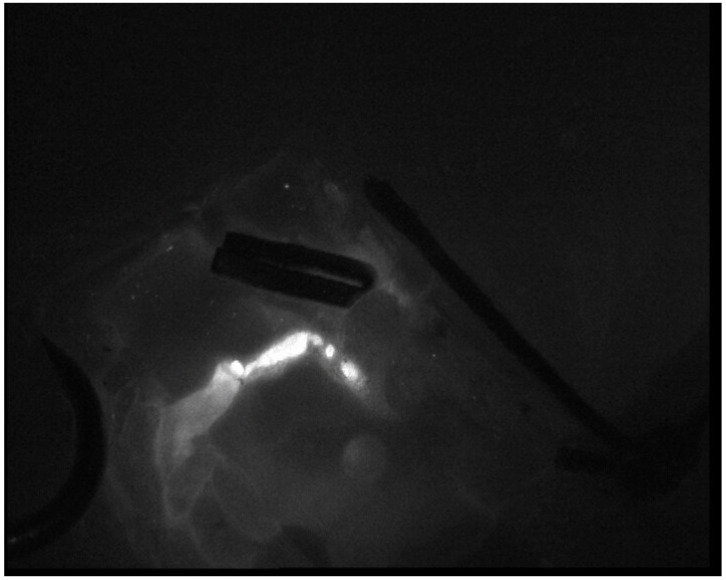
Confirmation of patency of lymphaticovenous anastomosis with ICG lymphangiography in Figure 1.

**Table 1 medicina-58-00207-t001:** Patients characteristics.

Case Number	Age (Years)	Sex	BMI	Pathology	Comorbidities
1	42	F	23	Breast cancer	None
2	51	F	30	Breast cancer	None
3	55	F	28	Breast cancer	None
4	48	F	24	Breast cancer	None
5	53	F	25	Breast cancer	None
6	60	F	21	Merkel Cell carcinoma	None

**Table 2 medicina-58-00207-t002:** Operative details, postoperative follow-up.

Case Number	Lymph Node Dissection	Time between Lymph Node Dissection and LVA (Days)	Number of Anastomosis	Length of Stay (Days)	Complications
1	Axillary	90	4	2	None
2	Axillary	110	4	2	None
3	Axillary	85	4	2	None
4	Axillary	120	4	2	None
5	Axillary	130	4	3	None
6	Inguinal	107	3	3	None

## Data Availability

Not applicable.

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
