# Peer review of "Distally Prophylactic Lymphaticovenular Anastomoses after Axillary or Inguinal Complete Lymph Node Dissection Followed by Radiotherapy: A Case Series"

_medicina, 2022, doi:10.3390/medicina58020207_

Round 1
Reviewer 1 Report
Nice case series, results are promising.
Author Response
We thank the reviewer for the thorough reading of the manuscript and his/her positive remarks.
Reviewer 2 Report
The reviewer agrees completely with the authors on the concept of prophylactic LVA. The manuscript is written concisely.
- The reviewer recommends for the authors to add some preoperative and postoperative information on ICG lymphography or lymphoscintigraphy.
- The reviewer recommends for the authors to add some limitation. Observation period in the study is short and the number of cases is small.
- In addition to limitation, the reviewer thinks additional discussion is necessary.
There is no consensus about the period of observation to determine if the prophylactic therapy is success or not, because there is so many factors to affect lymphatic function.
For example, one of them is aging, and the other one is gaining weight. It has been clarifying so far that lymphatic contraction force is decreased in old patients (1) and gaining weight can be the cause of lymphedema (2). Therefore, it is difficult to determine that prophylactic LVA was success in only a year. However, those articles report that LVA can improve lymphedema with those causes (2, 3). Therefore, the reviewer thinks prophylactic LVA is beneficial, but thinks difficult to determine that the prophylactic LVA was success. Lifetime observation is recommended for prophylactic LVA.
Please consider these suggestions and cite the recommended articles to make the article better one.
- Yoshida S. Indocyanine green lymphography findings in older patients with lower limb lymphedema. J Vasc Surg Venous Lymphat Disord. 2020 Mar;8(2):251-258.
- Yoshida S. Lymphovenous Anastomosis for Morbidly Obese Patients with Lymphedema. Plast Reconstr Surg Glob Open
. 2020 May 27;8(5):e2860.
- Yoshida S. Lymphaticovenous Anastomosis for Age-Related Lymphedema. J Clin Med. 2021 Oct 31;10(21):5129
Author Response
Reply to comments of the reviewer
A point-by-point answer has been provided. The comments of the reviewer are in italics
Reviewer #2
We thank the reviewer for the thorough reading of the manuscript and and his/her positive remarks. During the revision we had the chance to clarify some concepts.
The reviewer recommends for the authors to add some preoperative and postoperative information on ICG lymphography or lymphoscintigraphy.
- We use ICG lymphangiography to map the lymphatic ducts before making skin incisions and then after performing the anastomosis to confirm their intraoperative patency. A total of 2 cc of ICG was injected into the second and third interdigital space. We added these informations in the manuscript.
The reviewer recommends for the authors to add some limitation. Observation period in the study is short and the number of cases is small.
- We added these informations in the manuscript. We agree that a longer follow-up and a larger sample are required to determine the efficacy of our approach of prophylactic LVA.
In addition to limitation, the reviewer thinks additional discussion is necessary. There is no consensus about the period of observation to determine if the prophylactic therapy is success or not, because there is so many factors to affect lymphatic function. For example, one of them is aging, and the other one is gaining weight. It has been clarifying so far that lymphatic contraction force is decreased in old patients (1) and gaining weight can be the cause of lymphedema (2). Therefore, it is difficult to determine that prophylactic LVA was success in only a year. However, those articles report that LVA can improve lymphedema with those causes (2, 3). Therefore, the reviewer thinks prophylactic LVA is beneficial, but thinks difficult to determine that the prophylactic LVA was success. Lifetime observation is recommended for prophylactic LVA.
Please consider these suggestions and cite the recommended articles to make the article better one. Yoshida S. Indocyanine green lymphography findings in older patients with lower limb lymphedema. J Vasc Surg Venous Lymphat Disord. 2020 Mar;8(2):251-258.
Yoshida S. Lymphovenous Anastomosis for Morbidly Obese Patients with Lymphedema. Plast Reconstr Surg Glob Open
. 2020 May 27;8(5):e2860.
Yoshida S. Lymphaticovenous Anastomosis for Age-Related Lymphedema. J Clin Med. 2021 Oct 31;10(21):5129
- Thank you for the valuable comment. We added to discussion these suggestions and we mentioned previous articles in manuscript.
Reviewer 3 Report
1- in the Abstract, please add the number of patients included in the study
2- Add this citation and discuss it in the Introduction/Discussion:
Forte AJ, Sisti A, Huayllani MT, Boczar D, Cinotto G, Ciudad P, Manrique OJ,
Lu X, McLaughlin S. Lymphaticovenular anastomosis for breast cancer-related
upper extremity lymphedema: a literature review. Gland Surg. 2020
Apr;9(2):539-544. doi: 10.21037/gs.2020.03.41. PMID: 32420289; PMCID:
PMC7225471.
3- Change the title to:
prophylactic lymphaticovenular anastomoses in a distal site to axillary or groin region after axillary or inguinal complete lymph node dissection followed by radiotherapy: a case series
4- Add this reference to the Introduction:
Sisti A, Huayllani MT, Boczar D, Restrepo DJ, Spaulding AC, Emmanuel G,
Bagaria SP, McLaughlin SA, Parker AS, Forte AJ. Breast cancer in women: a
descriptive analysis of the national cancer database. Acta Biomed. 2020 May
11;91(2):332-341. doi: 10.23750/abm.v91i2.8399. PMID: 32420970; PMCID:
PMC7569667.
Author Response
Reply to comments of the reviewers
A point-by-point answer has been provided. The comments of the reviewer are in italics
Reviewer #3
We thank the reviewer for the thorough reading of the manuscript and his/her positive remarks. During the revision we had the chance to clarify some concepts. The manuscript was further edited.
In the Abstract, please add the number of patients included in the study.
- We thank the reviewer for the comment. We added the number of patients included in the study in the abstract.
Add this citation and discuss it in the Introduction/Discussion: Forte AJ, Sisti A, Huayllani MT, Boczar D, Cinotto G, Ciudad P, Manrique OJ, Lu X, McLaughlin S. Lymphaticovenular anastomosis for breast cancer-related upper extremity lymphedema: a literature review. Gland Surg. 2020 Apr;9(2):539-544. doi: 10.21037/gs.2020.03.41. PMID: 32420289; PMCID:
PMC7225471.
- We thank the reviewer for the suggestion.
Change the title to: prophylactic lymphaticovenular anastomoses in a distal site to axillary or groin region after axillary or inguinal complete lymph node dissection followed by radiotherapy: a case series.
- We thank for the suggestion. The title has now been changed to: “Distally prophylactic lymphaticovenular anastomoses after axillary or inguinal complete lymph node dissection followed by radiotherapy: a case series”.
Add this reference to the Introduction: Sisti A, Huayllani MT, Boczar D, Restrepo DJ, Spaulding AC, Emmanuel G, Bagaria SP, McLaughlin SA, Parker AS, Forte AJ. Breast cancer in women: a descriptive analysis of the national cancer database. Acta Biomed. 2020 May 11;91(2):332-341. doi: 10.23750/abm.v91i2.8399. PMID: 32420970; PMCID: PMC7569667.
- We added this reference to the introduction.
Round 2
Reviewer 2 Report
I am satisfied with the author's revision. I think the manuscript is acceptable for publication.
Author Response

(The authors gave the same response as above.)

Reviewer 3 Report
Thank you for your revisions. The manuscript requires some improvements. Please cite the 2 following articles:
1: Gillespie TC, Sayegh HE, Brunelle CL, Daniell KM, Taghian AG. Breast cancer- related lymphedema: risk factors, precautionary measures, and treatments. Gland Surg. 2018 Aug;7(4):379-403. doi: 10.21037/gs.2017.11.04. PMID: 30175055; PMCID:
PMC6107585.
2: Cook JA, Sinha M, Lester M, Fisher CS, Sen CK, Hassanein AH. Immediate
Lymphatic Reconstruction to Prevent Breast Cancer Related Lymphedema: A
Systematic Review. Adv Wound Care (New Rochelle). 2021 Oct 29. doi:
10.1089/wound.2021.0056. Epub ahead of print. PMID: 34714158.
and Discuss them. Do not only cite them, but discuss them in the Introduction/Discussion Section.
In particular, please discuss: Immediate lymphatic reconstruction (ILR), also termed Lymphatic Microsurgical Preventing Healing Approach (LyMPHA), which is a method to decrease the risk of lymphedema by performing prophylactic lymphovenous anastomoses at the time of ALND (axillary lymph node dissection).
Author Response
A point-by-point answer has been provided. The comment of the reviewer is in italics
Reviewer #3
We thank the reviewer for the thorough reading of the manuscript and and his/her positive remarks. During the revision we had the chance to clarify some concepts. The manuscript was further edited.
Please cite the 2 following articles:
1: Gillespie TC, Sayegh HE, Brunelle CL, Daniell KM, Taghian AG. Breast cancer- related lymphedema: risk factors, precautionary measures, and treatments. Gland Surg. 2018 Aug;7(4):379-403. doi: 10.21037/gs.2017.11.04. PMID: 30175055; PMCID:
PMC6107585.
2: Cook JA, Sinha M, Lester M, Fisher CS, Sen CK, Hassanein AH. Immediate
Lymphatic Reconstruction to Prevent Breast Cancer Related Lymphedema: A
Systematic Review. Adv Wound Care (New Rochelle). 2021 Oct 29. doi:
10.1089/wound.2021.0056. Epub ahead of print. PMID: 34714158.
- Thank you for the valuable comment. We mentioned previous articles in manuscript and we added to discussion these suggestions.